# New Amphiphilic Ionic Liquids for the Demulsification of Water-in-Heavy Crude Oil Emulsion

**DOI:** 10.3390/molecules27103238

**Published:** 2022-05-18

**Authors:** Mahmood M. S. Abdullah, Abdelrahman O. Ezzat, Hamad A. Al-Lohedan, Ali Aldalbahi, Ayman M. Atta

**Affiliations:** 1Department of Chemistry, College of Science, King Saud University, P.O. Box 2455, Riyadh 11451, Saudi Arabia; maltaiar@ksu.edu.sa (M.M.S.A.); hlohedan@ksu.edu.sa (H.A.A.-L.); aaldalbahi@ksu.edu.sa (A.A.); 2Petroleum Application Department, Egyptian Petroleum Research Institute, Cairo 11727, Egypt

**Keywords:** abietic acid, amphiphilic ionic liquids, crude oil emulsion, demulsification

## Abstract

This work aimed to use abietic acid (AA), as a widely available natural product, as a precursor for the synthesis of two new amphiphilic ionic liquids (AILs) and apply them as effective demulsifiers for water-in-crude oil (W/O) emulsions. AA was esterified using tetraethylene glycol (TEG) in the presence of *p*-toluene sulfonic acid (PTSA) as a catalyst obtaining the corresponding ester (AATG). AATG was reacted with 1-vinylimidazole (VIM) throughout the Diels–Alder reaction, forming the corresponding adduct (ATI). Following this, ATI was quaternized using alkyl iodides, ethyl iodide (EI), and hexyl iodide (HI) to obtain the corresponding AILs, ATEI-IL, and ATHI-IL, respectively. The chemical structure, surface activity, thermal stability, and relative solubility number (RSN) were investigated using different techniques. The efficiency of ATEI-IL and ATHI-IL to demulsify W/O emulsions in different crude oil: brine volumetric ratios were evaluated. ATEI-IL and ATHI-IL achieved promising results as demulsifiers. Their demulsification efficiency increased as the brine ratios decreased where their efficiency reached 100% at the crude oil: brine ratio (90:10), even at low concentrations.

## 1. Introduction

The presence of natural emulsifiers, such as asphaltenes, resins, and solid particles, as constituents in crude oil composition aids in the formation of stable crude oil emulsions with formation water during crude oil production [1,2]. In addition, the added amphiphilic chemicals such as surfactants during enhanced oil recovery, along with a high shear rate at reservoir conditions (high pressure and high temperature), enhances the stability of the formed crude oil emulsions [2,3]. The presence of crude oil in emulsion form leads to several operational issues, e.g., an increased crude oil viscosity, making its transportation very difficult, the corrosion of different equipment such as tanks, pipelines, and pumps, as well as the poisoning of catalysts during the refining process due to the presence of salts in the formation water. For that, the emulsified water should be separated before transportation and refining [4,5]. Chemical, physical, and biological techniques have been applied for the demulsification of crude oil emulsions [6,7]. The use of chemical demulsifiers is one of the most applied techniques due to their fast demulsification and low cost [8]. Surfactants are commonly used for the chemical demulsification of crude oil emulsions due to their amphiphilic character [9]. However, the efficiency of surfactants at harsh conditions such as high salinity, high pressure, and temperature is reduced significantly [10,11]. Over the past few years, several advanced materials have been proposed as demulsifiers for crude oil emulsions, such as reused demulsifiers [12], super-hydrophobic chemicals [13], and water-in-oil (W/O) emulsion inhibitors [14]. Recently, amphiphilic ionic liquids (ILs) were reported as effective demulsifiers for crude oil emulsions [15,16,17,18,19]. ILs are organic salts with low vapor pressure, high thermal stability, a low melting point (below 100 °C), and low toxicity. AILs have a great tendency to reduce the interfacial tension between crude oil and water even in harsh conditions; they disperse the asphaltenes’ interfacial film surrounding the water droplets, leading to phase separation [3,20].

Abietic acid (AA) is a natural acid extracted from *Pimeta recemosa* var. *Grissea*, where it represents the major component [21]. Due to its wide availability and green character, AA is used in different medical applications such as curing infected wounds, boils, and pyodermas [22], as well as for anti-inflammatory purposes. The complexes of AA with silver have shown good antibacterial activities for different types of bacteria [22,23]. The variations of functional groups, including a carboxylic group, double bonds, and a tricyclic in its structure, prompt its modification for the preparation of many derivatives. For that, AA has been used for the synthesis of different functional polymers, e.g., photosensitive and photo-crosslinked polymers, and sensitive adhesives [24]. In oilfield applications, in particular, AA has been used for the preparation of some esters and amides. The effect of the as-prepared esters and amides on the calorific value and crude oil flow was investigated [25]. In another study, AA-based amidoximes were used for the surface modification of magnetite nanoparticles and used for oil spill remediation [26]. Herein, two new AILs were synthesized based on the esterification of AA using tetraethylene glycol (TEG). The obtained ester was reacted with 1-vinylimidazole (VIM) throughout the Diels–Alder reaction. Following this, the obtained adduct was quaternized using ethyl iodide and hexyl iodide, obtaining the corresponding AILs, ATEI-IL, and ATHI-IL, respectively. The chemical structures, surface activity, and thermal stability of ATEI-IL, and ATHI-IL were investigated using different techniques. In addition, their efficiency for the demulsification of heavy crude oil emulsions was also evaluated.

## 2. Results and Discussion

In the present work, abietic acid as a natural product was esterified using tetraethylene glycol to introduce a hydrophilic moiety to its structure. After that, it was utilized to synthesize two amphiphilic ionic liquids through the Diels–Alder addition of vinyl imidazole, followed by alkylation on pyridine nitrogen using two alkyl iodides to form two new imidazolium ionic liquids. The prepared ionic liquids were applied successfully as demulsifiers for different water-in-oil emulsions.

### 2.1. Characterization of ILs and PILs

The chemical structures of the synthesized ILs confirmed using FTIR and ^1^HNMR spectra are shown in Figure 1 and Figure 2, respectively. The FTIR spectra of ATI, ATHI-IL and ATEI-IL are depicted in Figure 1a–c, respectively. The alcoholic hydroxyl group in all the prepared samples appeared as broad peaks at 3300–3400 cm^−1^. Additionally, the stretching bands at 2959 cm^−1^ and 2872 cm^−1^ are related to the aliphatic C-H, while the band at 1660 cm^−1^ is related to the stretching of C=CH. The shifting in carbonyl stretching bands from 1696 cm^−1^ for pure abietic acid to 1720 cm^−1^ for ATI, ATHI-IL and ATEI-IL verifies the esterification of abietic acid with tetraethylene glycol. The appearance of the stretching bands of aromatic C=C at 1569 cm^−1^ and 1652 cm^−1^ for both ATHI-IL and ATEI-IL confirmed the interaction of vinyl imidazole with abietic acid through the Diels–Alder reaction.

The ^1^HNMR spectra of ATI, ATHI-IL and ATEI-IL are shown in Figure 2a–c, respectively. Figure 2a confirms the esterification of abietic acid with tetraethylene glycol by the appearance of a triplet peak at 3.42–3.56 ppm related to (-CH_2_CH_2_O) and the appearance of the -OH broad peak at 5.3 ppm. Moreover, the appearance of doublet peaks related to 3H protons of imidazole at 7.57 and 7.92 ppm in ATI spectra (Figure 2a) confirms the addition of vinyl imidazole to the abietic ester. The shifting of imidazole protons and the appearance of a peak at 9.45 ppm for N-CHN^+^ and the appearance of a new peak at 4.3 ppm for CH_2_-N^+^ confirm the formation of imidazolium ionic liquids in both ATHI-IL and ATEI-IL, as indicated from Figure 2b,c, respectively. Other peaks are indicated on the chemical structures of ATEI-IL, as illustrated in Figure 2c.

Ionic liquid compounds are known for their high thermal stabilities that broaden their applications as green solvents, additives in enhanced oil recovery, etc. [27]. TGA-DTA measurements were used to investigate the thermal stabilities of the prepared ionic liquids (ATHI-IL and ATEI-IL), as shown in Figure 3a,b. The onset degradation temperatures for ATHI-IL and ATEI-IL were 280 °C and 274 °C, respectively, to verify their reasonable thermal stabilities. This small difference in the onset temperature can be regarded as the difference in the alky chain lengths between ATHI-IL and ATEI-IL that are ethyl and hexyl, respectively. Additionally, both ATHI-IL and ATEI-IL had a single decomposition step, as depicted in Figure 3a,b.

### 2.2. Solubility and Surface Activity of ATHI-IL and ATEI-IL

It has been mentioned that amphiphilic ionic liquids act as cationic surfactants with a greater tendency to form aggregates in their aqueous solutions [28]. The surface tension of different concentrations of the prepared ionic liquids in aqueous solutions was measured using the pendant drop method to study their behaviors. The surface tension values (γ) (mN/m) were plotted against ionic liquid concentrations (ln c) at constant temperatures (T), as shown in Figure 3, in order to determine the critical micelle concentrations (cmc; mmolL^−1^), surface excess concentration, Γ_max_, and the minimum area per molecule, A_min_, of ATHI-IL and ATEI-IL. The cmc was calculated from the intersection between the regression straight line of the linearly dependent region and the straight line passing through the plateau (Figure 4); Γ_max_ and A_min_ at the aqueous/air interface were calculated using the following Equations (1) and (2), respectively:
Γ_max_ = (−∂ γ/∂ ln c)_T_/RT(1)
A_min_ = 10^16^/N Γ_max_(2)
where (−∂γ/∂ ln c)_T_ is the slope of the plot, R is the gas constant (in J mol^−1^ K^−1^) and N is Avogadro’s number [29]. The calculated values for cmc, Γ_max_, A_min_ and (−∂ γ/∂ ln c) are tabulated in Table 1. As depicted in the table, the cmc values for ATHI-IL and ATEI-IL were 0.051 and 0.06 mM, respectively. The observed decrease in the cmc value for ATHI-IL can be attributed to the increment in the alkyl chain length, as stated in previous studies [30]. Additionally, according to previous studies, the more hydrophobic the ionic liquid is, the greater tendency to decrease the water surface tension [31]; this was obvious from the obtained results that showed the greater ability of ATHI-IL to decrease water surface tension than ATEI-IL. Furthermore, the higher hydrophobicity of ATHI-IL with hexyl alkyl chain led to a tighter packing of its molecules at the air/water interface, and this was confirmed from its higher value of Γ_max_ and lower one for A_min_, compared with ATEI-IL [32]. This also can be referred to the reduction in the electrostatic repulsion between pyridinium ions in ATHI-IL that led to more packing at the interface, lowering the cmc value [33]. The affinity of amphiphilic compounds to a given phase can be investigated practically using relative solubility number (RSN) measurements as an alternative method to the theoretical one (hydrophlic–lipophilic balance). The demulsifier tended to be soluble in water when the value of RSN exceeded 17, and the water solubility decreased as the RSN value decreased. The RSN values for ATHI-IL and ATEI-IL were 15.4 and 16.1, respectively, as listed in Table 1, indicating the low water solubility of the prepared ionic liquids.

### 2.3. Demulsification Efficiency of ATEI-IL and ATHI-IL

Nonionic surfactants are one of the most widely used surfactants in the petroleum industry due to their high ability to reduce interfacial tension (IFT) between crude oil and brine [34]. However, their interfacial activity is reduced significantly under harsh salinity conditions [35]. AILs contain hydrophobic and hydrophilic moieties, which means they can behave and work as nonionic surfactants. In addition, AILs have ions which means they can serve as cationic surfactants. Due to their structures, AILs show high performance even in high-salinity conditions. Ions of salt neutralize the opposite charge of AIL ions which reduces the repulsion between AIL molecules at the water/oil interface, resulting in the accumulation of AIL molecules at this interface and thereby leading to more IFT reduction [36]. The surface activity and RSN values of ATEI-IL and ATHI-IL confirm their amphiphilicity. For that, their demulsification efficiency (DE) using different concentrations was evaluated using the traditional bottle test, as mentioned in the Experimental section. Several volumetric ratios of crude oil: brine (50:50, 70:30, and 90:10) were used. The stability of the as-prepared emulsions was confirmed by placing the blank samples at 60 °C for a couple of weeks. The blank samples did not show any separation during this period confirming the stability of the as-prepared emulsions. The small size of the emulsion droplets (volumetric ratio 90:10) after two weeks with an average diameter of 2 µm (Figure 5a) also confirmed the formation of a stable emulsion.

Figure 5b,c show the optical microscopic images of emulsion droplets (volumetric ratio 90:10) using 500 ppm of ATEI-IL during the demulsification process. These images showed that the size of the water droplets increased with time. The diffusion of AIL in the continuous phase was followed by the adsorption of AIL molecules at the water/oil interface resulting in IFT reduction and thereby enhancing the replacement of the asphaltenes’ interfacial rigid film around the water droplets with a soft film, which facilitated the coalescence of droplets to form bigger droplets that settled at the cylinder’s bottom by gravity.

Demulsification efficiency and demulsification times are presented in Table 2 and Figure 6. Notably, the data show that the DE for both AILs reached 100% in different cases. The DE increased as the brine ratios in the as-prepared emulsions decreased. As a result, at a volumetric ratio of 90:10, the DE of ATEI-IL and ATHI-IL reached 100% even at low concentrations. By increasing the ratio of brine (volumetric ratio 50:50), the DE decreased significantly. These results may be associated with an increase in the hydrophobicity of ATEI-IL and ATHI-IL due to the presence of abietate rings that obstruct AIL diffusion with higher brine ratios. In addition, the DE increased as the AIL concentration increased. By increasing the AIL concentration, the number of adsorbed AIL molecules at the water/oil interface increased, leading to more IFT reduction and thereby replacing the asphaltene interfacial film around water droplets. At the volumetric ratio of crude oil: brine 70:30, the DE of ATHI-IL decreased as the concentration increased (from 500 ppm to 1000 ppm). This can be attributed to the aggregation of elevated concentrations of IL molecules at the water/oil interface to form new ATHI-IL films around the water droplets, leading to more emulsification instead of demulsification [37,38].

As it was reported earlier, the demulsification using amphiphilic compounds takes place in two main steps: diffusion and adsorption. In the first step, AILs diffuse in a crude oil as a continuous phase. An increase in the hydrophobicity of ATEI-IL and ATHI-IL due to the presence of abietate rings facilitates their diffusion in the crude oil as a continuous phase, and thus prompts their arrival to the water/oil interface [39,40]. In the second step, AIL molecules adsorb on oil/brine interfaces through a variety of interactions, including interactions between the hydrophilic AIL moiety (heteroatoms, e.g., the nitrogen and oxygen of AIL molecules) and the polar crude oil molecules, as well as interactions between the hydrophobic abietate rings of AIL molecules and the alkyl chains in crude oil [11]. AIL displayed a different type of interaction from traditional surfactant, such as ionic interaction between the AIL ions and the opposite charge of the salt ions in brine. These interactions reduced the repulsion between the molecules of AIL at the W/O interface, allowing them to accumulate at the interface. Due to that accumulation, more AIL molecules were available at the interface, leading to higher decrease in the IFT, and weaker asphaltene film, making it easier to replace and, therefore, lowering the stability of the emulsion droplets [16].

The difference in the chemical structures of ATEI-IL and ATHI-IL that arose from using two alkyl iodides with two different lengths exhibited a limited effect on their DE. However, the effect could be seen on the demulsification time, where ATEI-IL exhibited a relatively shorter demulsification time compared to that of ATHI-IL at all crude oil: brine volumetric ratios.

The ability of a demulsifier to separate clean water with no/low crude oil residual is one of the most important parameters for its selection due to environmental issues, as the presence of crude oil in the separated water requires further treatment before discharge. Figure 7a,b present the optical images of separated water in W/O emulsions (volumetric ratio 90:10) using different concentrations of ATEI-IL and ATHI-IL. These images exhibit the ability of ATEI-IL and ATHI-IL to separate clean water. Finally, the accumulative results indicate the high performance of the as-synthesized AILs to demulsify W/O emulsion, even at low concentrations. These results suggest the possibility of using ATEI-IL and ATHI-IL as commercial demulsifiers where they achieved high demulsification performance as single materials, as compared to other commercial demulsifiers which are usually a mixture of materials.

## 3. Materials and Methods

### 3.1. Materials

Abietic acid (AA ˃ 80%) was supplied by Tokyo Chemical Industry Co. Tetraethylene glycol (TEG ˃ 99%), 1-vinylimidazole (VIM ≥ 99), *p*-toluene sulfonic acid (PTSA ≥ 99), hydroquinone (≥99), ethyl iodide (EI 99%), hexyl iodide (HI ≥ 98%), toluene (≥99.7), dioxane (≥99), and xylene (≥98.5) were supplied by Sigma-Aldrich Co (Louis, MO, USA). Heavy crude oil was supplied by Aramco Co, Riyadh, Saudi Arabia. Its full specification is reported in our earlier work [18]. Brine (35,000 ppm) was prepared in our laboratory using distilled water and sodium chloride.

### 3.2. Synthesis of Amphiphilic Ionic Liquids

In a 100 mL two-neck round-bottom flask equipped with a magnetic stirrer and a nitrogen inlet connected to Dean-Starch apparatus, AA (5 g, 16.53 mmol), TEG (3.21 g, 16.53 mmol), and PTSA (0.14 g, 0.82 mmol) were dissolved in 25 mL of xylene. The mixture was refluxed for 5 h, followed by the evaporation of xylene under reduced pressure to produce the corresponding ester (AATG). Following this, AATG (10 g, 20.9 mmol) was heated at 180 in a three-neck round-bottom flask equipped with a magnetic stirrer, thermometer, condenser, and nitrogen inlet. VIM (1.97 g, 20.9 mmol) and hydroquinone (0.19 g) were added and stirred for 3 h at the same temperature; then, the temperature was raised to 220 °C and maintained for a further 1 h. When the temperature of the mixture reached 25 °C, hydroquinone was extracted using distilled water after dissolving the obtained mixture in chloroform. Chloroform was evaporated under reduced pressure to obtain the corresponding adduct ATI.

For the synthesis of the AILs, the equimolar of ATI with either EI or HI was dissolved in a certain amount of toluene and refluxed for 5 h. Toluene was evaporated under reduced pressure to obtain AILs. The obtained AILs using EI and HI were assigned as ATEI-IL and ATHI-IL, respectively. The synthesis route of ATEI-IL and ATHI-IL is presented in Figure 1.

### 3.3. Bottle Test Method

The demulsification efficiency of ATEI-IL and ATHI-IL was evaluated using the traditional bottle test method as follows: several ratios of crude oil: brine (50:50, 70:30, and 90:10) were mixed in a certain beaker using a homogenizer (BULLIO SR2) at 5000 rpm for 20 min at ambient temperature. The solutions of ATEI-IL and ATHI-IL were prepared by dissolving 0.5 g of AIL in 2 mL of xylene:ethanol (75:25 volumetric ratio). The as-prepared crude oil emulsion was transferred to 25 mL quick-fit cylinders and injected with the required concentration of AIL using a micropipette. The cylinders were closed, shaken 100 times to ensure the distribution of AIL in the prepared emulsion, and then placed in a hot water bath at 60 °C. When the AIL dose was injected into the emulsion, the time was recorded as zero to measure the demulsification time. Blank samples were treated in the same way except that they were not injected with AILs. The blank samples were injected with the same amount of xylene:ethanol, free of AIL. The demulsification efficiency (DE%) of ATEI-IL and ATHI-IL was calculated using the following Equation (3):(3)DE%=SWEW×100
where SW and EW are the volume of separated water after the demulsification process and the volume of emulsified water during the preparation of emulsion, respectively.

### 3.4. Characterization

Fourier transform infrared (FTIR, Nicolet 6700 spectrometer, Waltham, MA, USA) spectroscopy was used to verify the chemical structures of ATEI-IL and ATHI-IL. For that, AIL samples were ground with KBr powder (spectroscopic grade) and passed into a 1 mm pellet for measurements in the range of wavenumber 4000 cm^−1^–400 cm^−1^. Proton nuclear magnetic resonance (^1^H-NMR, Avance DRX-400 spectrometer, Billerica, MA, USA) spectroscopy was also used to confirm the chemical structures of ATEI-IL and ATHI-IL. To do so, AIL samples were dissolved in deuterated dimethyl sulfoxide-*d6* (DMSO-*d6*). The thermal stability of ATEI-IL and ATHI-IL was determined via thermal gravimetric analysis (TGA) and differential thermal analysis (DTG) (Shimadzu, DSC-60 Instrument, New York, NY, USA) while heating the AIL in a range of 25–800 °C under a nitrogen atmosphere at 10 C/min. The pendant drop technique using a drop shape analyzer (DSA-100, Kruss Hamburg, Germany) was used to investigate the surface tension of ATEI-IL and ATHI-IL at the air/water interface. Relative solubility number (RSN) was used to investigate the amphiphilicity and solubility of the as-synthesized AILs in different solvents. RSN was defined as the volume of distilled water (in mL) required for treating the AIL solution (1 g of AIL in 30 mL of dioxane: toluene 94:6 volumetric ratio) until the appearance of continuous turbidity. Polarized light microscopy (Olympus BX51, Tokyo, Japan) was used for capturing microscopic photos of emulsions before and during the demulsification process.

## 4. Conclusions

This work aimed to use AA, as a widely available natural compound, to synthesize new AILs and apply them for the demulsification of W/O emulsions. The reaction between AA and TEG produced the corresponding ester AATG. AILs, ATEI-IL, and ATHI-IL were obtained by reacting the produced ester with VIM throughout the Diels–Alder reaction, then quaternizing the obtained adduct, ATI, with either EI or HI, respectively. The chemical structures, surface activity, thermal stability, and RSN were investigated using different techniques. The surface activity parameters and RSN confirmed the amphiphilicity of ATEI-IL and ATHI-IL.

Thanks to the amphiphilicity of the as-synthesized AILs, their performance for the demulsification of W/O emulsions with different crude oil: brine volumetric ratios was evaluated. At the crude oil: brine volumetric ratio of 90:10, the DE for both AILs reached 100% at all concentrations, which may be referred to as increasing their hydrophobicity due to the presence of abietate rings. Increasing the hydrophobicity of AIL facilitates their diffusion in the crude oil as a continuous phase and thus prompts their arrival at the water/oil interface. The adsorption of AILs at the W/O interface is enhanced via a verity of interactions, including interactions between the hydrophilic AIL moiety (heteroatoms, e.g., the nitrogen and oxygen of AIL molecules) and the polar crude oil molecules, as well as interactions between the hydrophobic abietate rings of the AIL molecules and the corresponding alkyl chains in crude oil. Such interactions reduce IFT and thereby replace the asphaltenes’ interfacial film. By increasing the brine ratio (volumetric ratio 50:50), the DE for both AILs declined significantly. This could be due to an increase in the hydrophobicity of ATEI-IL and ATHI-IL because of the presence of abietate rings that obstruct the AIL diffusion when increasing the brine ratio. In addition, the DE increased as the AIL concentration increased, which may be linked to an increase in the number of adsorbed AIL molecules at the water/oil interface with increasing AIL concentration, enhancing the IFT reduction. The optical microscopic images exhibited the fusion of small emulsion droplets forming bigger ones due to the replacement of the asphaltenes’ interfacial film with a soft film which facilitated the coalescence of droplets to form bigger droplets that settled at the cylinder’s bottom by gravity. The optical images of the separated water in the cylinders indicate the ability of the as-synthesized AILs to separate clean water. Finally, the green character and wide availability of AA makes it a suitable precursor for the synthesis of AILs, ATEI-IL, and ATHI-IL in a short synthesis route. In addition, the high demulsification performance, short demulsification time, and low injected doses of the as-synthesized AILs confirm the efficiency of these AILs to serve as effective commercial demulsifiers for W/O emulsions, especially with decreased brine ratios in these emulsions.

## Data Availability

The data presented in this study are available on request from the corresponding author.

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
