# Peer review of "New Amphiphilic Ionic Liquids for the Demulsification of Water-in-Heavy Crude Oil Emulsion"

_molecules, 2022, doi:10.3390/molecules27103238_

Round 1

Reviewer 1 Report

Two novel ionic liquids were synthesized and the demulsification performance in crude oil-water emulsions was evaluated in the manuscript. I recommend publication of this manuscript after major revisions. Some suggestions are also provided for improving the manuscript.

  1. What is the role of hydroquinone in the synthesis process of ATI?
  2. In scheme 1, what is the meaning of the double arrow? Although the molar ratio of TEG to AA is 1:1, is it possible for one TEG molecule to react with two AA molecules? In addition, RBr should be changed to RI.
  3. L81,“carbonyl starching band” should be changed to “carbonyl stretching bands”.
  4. Only the main characteristic peaks need to be marked in the IR spectra.
  5. The authors describe that ionic liquid compound has high thermal stabilities, so the TGA of non-ionic liquid ATI should be provided as a comparison.
  6. In table 2, why does the concentration of 1000mg lead to a lower demulsification efficiency compared with the concentration of 500ppm when the volumetric ratio of crude oil and brine is 70:30? In addition, only low demulsification efficiency can be obtained when the brine content is relatively high, the reason needs to be explained.
  7. ATHI-IL has a greater ability to decrease the water surface tension than ATEI-IL. However, the demulsification ability of ATHI-IL is stronger than ATEI-IL. Is the result contradictory?
  8. Are ATHI-IL and ATEI-IL more efficient than non-ionic liquid ATI?
  9. The conclusion section needs to be further improved.

Author Response

Reviewer 1

Two novel ionic liquids were synthesized and the demulsification performance in crude oil-water emulsions was evaluated in the manuscript. I recommend publication of this manuscript after major revisions. Some suggestions are also provided for improving the manuscript.

  1. What is the role of hydroquinone in the synthesis process of ATI?

Answer: Hydroquinone is commonly used to inhibit the polymerization of monomers containing double bonds. During the Diels-Alder reaction, at high temperature, 1-vinylimidazole (VIM) monomers tend to react with each other forming a polymer. The addition of hydroquinone in this reaction inhibits the polymerization between 1-vinylimidazole (VIM).

  1. In scheme 1, what is the meaning of the double arrow? Although the molar ratio of TEG to AA is 1:1, is it possible for one TEG molecule to react with two AA molecules? In addition, RBr should be changed to RI.

Answer: the meaning of double arrow is that there is equilibrium between the two isomers of abietic acid at the temperature range from 180 to 220. the cis form is the suitable one for the Diels Alder addition.

By the 1HNMR analysis, the reaction of tetraethylene glycol from one size was confirmed by the appearance of a broad peak at chemical shift of 5.4, and also by the appearance of broad peak related to OH stretching vibration in the FTIR spectrum at peaks at 3300-3400 cm−1.

RBr was replaced by RI.

  1. L81,“carbonyl starching band” should be changed to “carbonyl stretching bands”.

Answer: It was modified to “carbonyl stretching bands”.

  1. Only the main characteristic peaks need to be marked in the IR spectra.

Answer: the main characteristic bands were pointer on the FTIR chart and other peaks were removed.

  1. The authors describe that ionic liquid compound has high thermal stabilities, so the TGA of non-ionic liquid ATI should be provided as a comparison.

Answer: As mentioned in discussion part, ‘’the onset degradation temperatures for ATHI-IL and ATEI-IL were 280 °C and 274 °C, that verify the reasonable thermal stabilities of the prepared materials. As we use the current prepared materials as demulsifiers at 60 °C, they are suitable for the current application. The thermal studied were carried out to indicate that the thermal stabilities of the prepared material are suitable even for application that require higher temperature. However, we sincerely apologize for the reviewer that we cannot perform the TGA analysis for non-ionic liquid ATI due to some technical problems with the instruments.

  1. In table 2, why does the concentration of 1000mg lead to a lower demulsification efficiency compared with the concentration of 500ppm when the volumetric ratio of crude oil and brine is 70:30? In addition, only low demulsification efficiency can be obtained when the brine content is relatively high, the reason needs to be explained.

Answer: More explanations were added to the manuscript to explain the above-mentioned reasons.

  1. ATHI-IL has a greater ability to decrease the water surface tension than ATEI-IL. However, the demulsification ability of ATHI-IL is stronger than ATEI-IL. Is the result contradictory?

Answer: As it can be seen in Table 2, while both AILs, ATEI-IL, and ATHI-IL achieved the same demulsification efficiency (100%) at the volumetric ratio 90:10, ATEI-IL displayed higher demulsification efficiency at volumetric ratios 70:30 and 50:50 as compared to ATHI-IL. In addition, ATEI-IL exhibited shorter demulsification time as compared to ATHI-IL.

  1. Are ATHI-IL and ATEI-IL more efficient than non-ionic liquid ATI?

Answer: Yes, ATHI-IL and ATEI-IL displayed higher efficiency than non-ionic liquid ATI.

  1. The conclusion section needs to be further improved.

Answer: The conclusion section was improved.

Reviewer 1

Two novel ionic liquids were synthesized and the demulsification performance in crude oil-water emulsions was evaluated in the manuscript. I recommend publication of this manuscript after major revisions. Some suggestions are also provided for improving the manuscript.

  1. What is the role of hydroquinone in the synthesis process of ATI?

Answer: Hydroquinone is commonly used to inhibit the polymerization of monomers containing double bonds. During the Diels-Alder reaction, at high temperature, 1-vinylimidazole (VIM) monomers tend to react with each other forming a polymer. The addition of hydroquinone in this reaction inhibits the polymerization between 1-vinylimidazole (VIM).

  1. In scheme 1, what is the meaning of the double arrow? Although the molar ratio of TEG to AA is 1:1, is it possible for one TEG molecule to react with two AA molecules? In addition, RBr should be changed to RI.

Answer: the meaning of double arrow is that there is equilibrium between the two isomers of abietic acid at the temperature range from 180 to 220. the cis form is the suitable one for the Diels Alder addition.

By the 1HNMR analysis, the reaction of tetraethylene glycol from one size was confirmed by the appearance of a broad peak at chemical shift of 5.4, and also by the appearance of broad peak related to OH stretching vibration in the FTIR spectrum at peaks at 3300-3400 cm−1.

RBr was replaced by RI.

  1. L81,“carbonyl starching band” should be changed to “carbonyl stretching bands”.

Answer: It was modified to “carbonyl stretching bands”.

  1. Only the main characteristic peaks need to be marked in the IR spectra.

Answer: the main characteristic bands were pointer on the FTIR chart and other peaks were removed.

  1. The authors describe that ionic liquid compound has high thermal stabilities, so the TGA of non-ionic liquid ATI should be provided as a comparison.

Answer: As mentioned in discussion part, ‘’the onset degradation temperatures for ATHI-IL and ATEI-IL were 280 °C and 274 °C, that verify the reasonable thermal stabilities of the prepared materials. As we use the current prepared materials as demulsifiers at 60 °C, they are suitable for the current application. The thermal studied were carried out to indicate that the thermal stabilities of the prepared material are suitable even for application that require higher temperature. However, we sincerely apologize for the reviewer that we cannot perform the TGA analysis for non-ionic liquid ATI due to some technical problems with the instruments.

  1. In table 2, why does the concentration of 1000mg lead to a lower demulsification efficiency compared with the concentration of 500ppm when the volumetric ratio of crude oil and brine is 70:30? In addition, only low demulsification efficiency can be obtained when the brine content is relatively high, the reason needs to be explained.

Answer: More explanations were added to the manuscript to explain the above-mentioned reasons.

  1. ATHI-IL has a greater ability to decrease the water surface tension than ATEI-IL. However, the demulsification ability of ATHI-IL is stronger than ATEI-IL. Is the result contradictory?

Answer: As it can be seen in Table 2, while both AILs, ATEI-IL, and ATHI-IL achieved the same demulsification efficiency (100%) at the volumetric ratio 90:10, ATEI-IL displayed higher demulsification efficiency at volumetric ratios 70:30 and 50:50 as compared to ATHI-IL. In addition, ATEI-IL exhibited shorter demulsification time as compared to ATHI-IL.

  1. Are ATHI-IL and ATEI-IL more efficient than non-ionic liquid ATI?

Answer: Yes, ATHI-IL and ATEI-IL displayed higher efficiency than non-ionic liquid ATI.

  1. The conclusion section needs to be further improved.

Answer: The conclusion section was improved.

Reviewer 1

Two novel ionic liquids were synthesized and the demulsification performance in crude oil-water emulsions was evaluated in the manuscript. I recommend publication of this manuscript after major revisions. Some suggestions are also provided for improving the manuscript.

  1. What is the role of hydroquinone in the synthesis process of ATI?

Answer: Hydroquinone is commonly used to inhibit the polymerization of monomers containing double bonds. During the Diels-Alder reaction, at high temperature, 1-vinylimidazole (VIM) monomers tend to react with each other forming a polymer. The addition of hydroquinone in this reaction inhibits the polymerization between 1-vinylimidazole (VIM).

  1. In scheme 1, what is the meaning of the double arrow? Although the molar ratio of TEG to AA is 1:1, is it possible for one TEG molecule to react with two AA molecules? In addition, RBr should be changed to RI.

Answer: the meaning of double arrow is that there is equilibrium between the two isomers of abietic acid at the temperature range from 180 to 220. the cis form is the suitable one for the Diels Alder addition.

By the 1HNMR analysis, the reaction of tetraethylene glycol from one size was confirmed by the appearance of a broad peak at chemical shift of 5.4, and also by the appearance of broad peak related to OH stretching vibration in the FTIR spectrum at peaks at 3300-3400 cm−1.

RBr was replaced by RI.

  1. L81,“carbonyl starching band” should be changed to “carbonyl stretching bands”.

Answer: It was modified to “carbonyl stretching bands”.

  1. Only the main characteristic peaks need to be marked in the IR spectra.

Answer: the main characteristic bands were pointer on the FTIR chart and other peaks were removed.

  1. The authors describe that ionic liquid compound has high thermal stabilities, so the TGA of non-ionic liquid ATI should be provided as a comparison.

Answer: As mentioned in discussion part, ‘’the onset degradation temperatures for ATHI-IL and ATEI-IL were 280 °C and 274 °C, that verify the reasonable thermal stabilities of the prepared materials. As we use the current prepared materials as demulsifiers at 60 °C, they are suitable for the current application. The thermal studied were carried out to indicate that the thermal stabilities of the prepared material are suitable even for application that require higher temperature. However, we sincerely apologize for the reviewer that we cannot perform the TGA analysis for non-ionic liquid ATI due to some technical problems with the instruments.

  1. In table 2, why does the concentration of 1000mg lead to a lower demulsification efficiency compared with the concentration of 500ppm when the volumetric ratio of crude oil and brine is 70:30? In addition, only low demulsification efficiency can be obtained when the brine content is relatively high, the reason needs to be explained.

Answer: More explanations were added to the manuscript to explain the above-mentioned reasons.

  1. ATHI-IL has a greater ability to decrease the water surface tension than ATEI-IL. However, the demulsification ability of ATHI-IL is stronger than ATEI-IL. Is the result contradictory?

Answer: As it can be seen in Table 2, while both AILs, ATEI-IL, and ATHI-IL achieved the same demulsification efficiency (100%) at the volumetric ratio 90:10, ATEI-IL displayed higher demulsification efficiency at volumetric ratios 70:30 and 50:50 as compared to ATHI-IL. In addition, ATEI-IL exhibited shorter demulsification time as compared to ATHI-IL.

  1. Are ATHI-IL and ATEI-IL more efficient than non-ionic liquid ATI?

Answer: Yes, ATHI-IL and ATEI-IL displayed higher efficiency than non-ionic liquid ATI.

  1. The conclusion section needs to be further improved.

Answer: The conclusion section was improved.

Reviewer 1

Two novel ionic liquids were synthesized and the demulsification performance in crude oil-water emulsions was evaluated in the manuscript. I recommend publication of this manuscript after major revisions. Some suggestions are also provided for improving the manuscript.

  1. What is the role of hydroquinone in the synthesis process of ATI?

Answer: Hydroquinone is commonly used to inhibit the polymerization of monomers containing double bonds. During the Diels-Alder reaction, at high temperature, 1-vinylimidazole (VIM) monomers tend to react with each other forming a polymer. The addition of hydroquinone in this reaction inhibits the polymerization between 1-vinylimidazole (VIM).

  1. In scheme 1, what is the meaning of the double arrow? Although the molar ratio of TEG to AA is 1:1, is it possible for one TEG molecule to react with two AA molecules? In addition, RBr should be changed to RI.

Answer: the meaning of double arrow is that there is equilibrium between the two isomers of abietic acid at the temperature range from 180 to 220. the cis form is the suitable one for the Diels Alder addition.

By the 1HNMR analysis, the reaction of tetraethylene glycol from one size was confirmed by the appearance of a broad peak at chemical shift of 5.4, and also by the appearance of broad peak related to OH stretching vibration in the FTIR spectrum at peaks at 3300-3400 cm−1.

RBr was replaced by RI.

  1. L81,“carbonyl starching band” should be changed to “carbonyl stretching bands”.

Answer: It was modified to “carbonyl stretching bands”.

  1. Only the main characteristic peaks need to be marked in the IR spectra.

Answer: the main characteristic bands were pointer on the FTIR chart and other peaks were removed.

  1. The authors describe that ionic liquid compound has high thermal stabilities, so the TGA of non-ionic liquid ATI should be provided as a comparison.

Answer: As mentioned in discussion part, ‘’the onset degradation temperatures for ATHI-IL and ATEI-IL were 280 °C and 274 °C, that verify the reasonable thermal stabilities of the prepared materials. As we use the current prepared materials as demulsifiers at 60 °C, they are suitable for the current application. The thermal studied were carried out to indicate that the thermal stabilities of the prepared material are suitable even for application that require higher temperature. However, we sincerely apologize for the reviewer that we cannot perform the TGA analysis for non-ionic liquid ATI due to some technical problems with the instruments.

  1. In table 2, why does the concentration of 1000mg lead to a lower demulsification efficiency compared with the concentration of 500ppm when the volumetric ratio of crude oil and brine is 70:30? In addition, only low demulsification efficiency can be obtained when the brine content is relatively high, the reason needs to be explained.

Answer: More explanations were added to the manuscript to explain the above-mentioned reasons.

  1. ATHI-IL has a greater ability to decrease the water surface tension than ATEI-IL. However, the demulsification ability of ATHI-IL is stronger than ATEI-IL. Is the result contradictory?

Answer: As it can be seen in Table 2, while both AILs, ATEI-IL, and ATHI-IL achieved the same demulsification efficiency (100%) at the volumetric ratio 90:10, ATEI-IL displayed higher demulsification efficiency at volumetric ratios 70:30 and 50:50 as compared to ATHI-IL. In addition, ATEI-IL exhibited shorter demulsification time as compared to ATHI-IL.

  1. Are ATHI-IL and ATEI-IL more efficient than non-ionic liquid ATI?

Answer: Yes, ATHI-IL and ATEI-IL displayed higher efficiency than non-ionic liquid ATI.

  1. The conclusion section needs to be further improved.

Answer: The conclusion section was improved.

Reviewer 1

Two novel ionic liquids were synthesized and the demulsification performance in crude oil-water emulsions was evaluated in the manuscript. I recommend publication of this manuscript after major revisions. Some suggestions are also provided for improving the manuscript.

  1. What is the role of hydroquinone in the synthesis process of ATI?

Answer: Hydroquinone is commonly used to inhibit the polymerization of monomers containing double bonds. During the Diels-Alder reaction, at high temperature, 1-vinylimidazole (VIM) monomers tend to react with each other forming a polymer. The addition of hydroquinone in this reaction inhibits the polymerization between 1-vinylimidazole (VIM).

  1. In scheme 1, what is the meaning of the double arrow? Although the molar ratio of TEG to AA is 1:1, is it possible for one TEG molecule to react with two AA molecules? In addition, RBr should be changed to RI.

Answer: the meaning of double arrow is that there is equilibrium between the two isomers of abietic acid at the temperature range from 180 to 220. the cis form is the suitable one for the Diels Alder addition.

By the 1HNMR analysis, the reaction of tetraethylene glycol from one size was confirmed by the appearance of a broad peak at chemical shift of 5.4, and also by the appearance of broad peak related to OH stretching vibration in the FTIR spectrum at peaks at 3300-3400 cm−1.

RBr was replaced by RI.

  1. L81,“carbonyl starching band” should be changed to “carbonyl stretching bands”.

Answer: It was modified to “carbonyl stretching bands”.

  1. Only the main characteristic peaks need to be marked in the IR spectra.

Answer: the main characteristic bands were pointer on the FTIR chart and other peaks were removed.

  1. The authors describe that ionic liquid compound has high thermal stabilities, so the TGA of non-ionic liquid ATI should be provided as a comparison.

Answer: As mentioned in discussion part, ‘’the onset degradation temperatures for ATHI-IL and ATEI-IL were 280 °C and 274 °C, that verify the reasonable thermal stabilities of the prepared materials. As we use the current prepared materials as demulsifiers at 60 °C, they are suitable for the current application. The thermal studied were carried out to indicate that the thermal stabilities of the prepared material are suitable even for application that require higher temperature. However, we sincerely apologize for the reviewer that we cannot perform the TGA analysis for non-ionic liquid ATI due to some technical problems with the instruments.

  1. In table 2, why does the concentration of 1000mg lead to a lower demulsification efficiency compared with the concentration of 500ppm when the volumetric ratio of crude oil and brine is 70:30? In addition, only low demulsification efficiency can be obtained when the brine content is relatively high, the reason needs to be explained.

Answer: More explanations were added to the manuscript to explain the above-mentioned reasons.

  1. ATHI-IL has a greater ability to decrease the water surface tension than ATEI-IL. However, the demulsification ability of ATHI-IL is stronger than ATEI-IL. Is the result contradictory?

Answer: As it can be seen in Table 2, while both AILs, ATEI-IL, and ATHI-IL achieved the same demulsification efficiency (100%) at the volumetric ratio 90:10, ATEI-IL displayed higher demulsification efficiency at volumetric ratios 70:30 and 50:50 as compared to ATHI-IL. In addition, ATEI-IL exhibited shorter demulsification time as compared to ATHI-IL.

  1. Are ATHI-IL and ATEI-IL more efficient than non-ionic liquid ATI?

Answer: Yes, ATHI-IL and ATEI-IL displayed higher efficiency than non-ionic liquid ATI.

  1. The conclusion section needs to be further improved.

Answer: The conclusion section was improved.

Reviewer 1

Two novel ionic liquids were synthesized and the demulsification performance in crude oil-water emulsions was evaluated in the manuscript. I recommend publication of this manuscript after major revisions. Some suggestions are also provided for improving the manuscript.

  1. What is the role of hydroquinone in the synthesis process of ATI?

Answer: Hydroquinone is commonly used to inhibit the polymerization of monomers containing double bonds. During the Diels-Alder reaction, at high temperature, 1-vinylimidazole (VIM) monomers tend to react with each other forming a polymer. The addition of hydroquinone in this reaction inhibits the polymerization between 1-vinylimidazole (VIM).

  1. In scheme 1, what is the meaning of the double arrow? Although the molar ratio of TEG to AA is 1:1, is it possible for one TEG molecule to react with two AA molecules? In addition, RBr should be changed to RI.

Answer: the meaning of double arrow is that there is equilibrium between the two isomers of abietic acid at the temperature range from 180 to 220. the cis form is the suitable one for the Diels Alder addition.

By the 1HNMR analysis, the reaction of tetraethylene glycol from one size was confirmed by the appearance of a broad peak at chemical shift of 5.4, and also by the appearance of broad peak related to OH stretching vibration in the FTIR spectrum at peaks at 3300-3400 cm−1.

RBr was replaced by RI.

  1. L81,“carbonyl starching band” should be changed to “carbonyl stretching bands”.

Answer: It was modified to “carbonyl stretching bands”.

  1. Only the main characteristic peaks need to be marked in the IR spectra.

Answer: the main characteristic bands were pointer on the FTIR chart and other peaks were removed.

  1. The authors describe that ionic liquid compound has high thermal stabilities, so the TGA of non-ionic liquid ATI should be provided as a comparison.

Answer: As mentioned in discussion part, ‘’the onset degradation temperatures for ATHI-IL and ATEI-IL were 280 °C and 274 °C, that verify the reasonable thermal stabilities of the prepared materials. As we use the current prepared materials as demulsifiers at 60 °C, they are suitable for the current application. The thermal studied were carried out to indicate that the thermal stabilities of the prepared material are suitable even for application that require higher temperature. However, we sincerely apologize for the reviewer that we cannot perform the TGA analysis for non-ionic liquid ATI due to some technical problems with the instruments.

  1. In table 2, why does the concentration of 1000mg lead to a lower demulsification efficiency compared with the concentration of 500ppm when the volumetric ratio of crude oil and brine is 70:30? In addition, only low demulsification efficiency can be obtained when the brine content is relatively high, the reason needs to be explained.

Answer: More explanations were added to the manuscript to explain the above-mentioned reasons.

  1. ATHI-IL has a greater ability to decrease the water surface tension than ATEI-IL. However, the demulsification ability of ATHI-IL is stronger than ATEI-IL. Is the result contradictory?

Answer: As it can be seen in Table 2, while both AILs, ATEI-IL, and ATHI-IL achieved the same demulsification efficiency (100%) at the volumetric ratio 90:10, ATEI-IL displayed higher demulsification efficiency at volumetric ratios 70:30 and 50:50 as compared to ATHI-IL. In addition, ATEI-IL exhibited shorter demulsification time as compared to ATHI-IL.

  1. Are ATHI-IL and ATEI-IL more efficient than non-ionic liquid ATI?

Answer: Yes, ATHI-IL and ATEI-IL displayed higher efficiency than non-ionic liquid ATI.

  1. The conclusion section needs to be further improved.

Answer: The conclusion section was improved.

Reviewer 1

Two novel ionic liquids were synthesized and the demulsification performance in crude oil-water emulsions was evaluated in the manuscript. I recommend publication of this manuscript after major revisions. Some suggestions are also provided for improving the manuscript.

  1. What is the role of hydroquinone in the synthesis process of ATI?

Answer: Hydroquinone is commonly used to inhibit the polymerization of monomers containing double bonds. During the Diels-Alder reaction, at high temperature, 1-vinylimidazole (VIM) monomers tend to react with each other forming a polymer. The addition of hydroquinone in this reaction inhibits the polymerization between 1-vinylimidazole (VIM).

  1. In scheme 1, what is the meaning of the double arrow? Although the molar ratio of TEG to AA is 1:1, is it possible for one TEG molecule to react with two AA molecules? In addition, RBr should be changed to RI.

Answer: the meaning of double arrow is that there is equilibrium between the two isomers of abietic acid at the temperature range from 180 to 220. the cis form is the suitable one for the Diels Alder addition.

By the 1HNMR analysis, the reaction of tetraethylene glycol from one size was confirmed by the appearance of a broad peak at chemical shift of 5.4, and also by the appearance of broad peak related to OH stretching vibration in the FTIR spectrum at peaks at 3300-3400 cm−1.

RBr was replaced by RI.

  1. L81,“carbonyl starching band” should be changed to “carbonyl stretching bands”.

Answer: It was modified to “carbonyl stretching bands”.

  1. Only the main characteristic peaks need to be marked in the IR spectra.

Answer: the main characteristic bands were pointer on the FTIR chart and other peaks were removed.

  1. The authors describe that ionic liquid compound has high thermal stabilities, so the TGA of non-ionic liquid ATI should be provided as a comparison.

Answer: As mentioned in discussion part, ‘’the onset degradation temperatures for ATHI-IL and ATEI-IL were 280 °C and 274 °C, that verify the reasonable thermal stabilities of the prepared materials. As we use the current prepared materials as demulsifiers at 60 °C, they are suitable for the current application. The thermal studied were carried out to indicate that the thermal stabilities of the prepared material are suitable even for application that require higher temperature. However, we sincerely apologize for the reviewer that we cannot perform the TGA analysis for non-ionic liquid ATI due to some technical problems with the instruments.

  1. In table 2, why does the concentration of 1000mg lead to a lower demulsification efficiency compared with the concentration of 500ppm when the volumetric ratio of crude oil and brine is 70:30? In addition, only low demulsification efficiency can be obtained when the brine content is relatively high, the reason needs to be explained.

Answer: More explanations were added to the manuscript to explain the above-mentioned reasons.

  1. ATHI-IL has a greater ability to decrease the water surface tension than ATEI-IL. However, the demulsification ability of ATHI-IL is stronger than ATEI-IL. Is the result contradictory?

Answer: As it can be seen in Table 2, while both AILs, ATEI-IL, and ATHI-IL achieved the same demulsification efficiency (100%) at the volumetric ratio 90:10, ATEI-IL displayed higher demulsification efficiency at volumetric ratios 70:30 and 50:50 as compared to ATHI-IL. In addition, ATEI-IL exhibited shorter demulsification time as compared to ATHI-IL.

  1. Are ATHI-IL and ATEI-IL more efficient than non-ionic liquid ATI?

Answer: Yes, ATHI-IL and ATEI-IL displayed higher efficiency than non-ionic liquid ATI.

  1. The conclusion section needs to be further improved.

Answer: The conclusion section was improved.

Reviewer 2 Report

The manuscript entitled “New Amphiphilic Ionic liquids for Demulsification of Water-2 in-Heavy Crude Oil Emulsion” by Mahmood M. S. Abdullah and co-workers presents the synthesis of 2 ionic liquids (ATEI-IL and ATHI-IL) and their application for chemical demulsification. To characterize the discussed system, the authors used several techniques, including FTIR and 1H-NMR spectroscopy, TGA and DTA analysis, surface tension measurement and optical microscopy.

I have the following comments:

  1. Page 2, line 81: Should be “carbonyl stretching band” instead of “carbonyl starching band”
  2. Figure 1b: there is no wavelength value for the CH2 antisymmetric stretching vibration.
  3. Page 5, line 119: the authors wrote that the CMC of ionic liquids was calculated “from the intersection between the regression straight line of the linearly dependent region and the straight line passing through the plateau” and indeed, it should be shown in Figure 4. Simple point joining is not appropriate in this case.
  4. Page 6, line 129: “that” instead of “That”.
  5. Figure 4: description of the x axis: should be “ln c”, instead of “lnC”
  6. Figure 6 is not referenced anywhere.
  7. The purity of the chemicals used should be reported.
  8. Scheme I, the last step of the synthesis: is “RBr” should be “RI”.
  9. Chapter 3.4: the experimental details are poorly described.
  10. Page 12: the first sentence of the conclusion must be corrected.
  11. The practical relevance of the conclusions needs to be addressed.

In my opinion, the manuscript is suitable for publication after making corrections.

Author Response

 Reviewer 2

The manuscript entitled “New Amphiphilic Ionic liquids for Demulsification of Water-2 in-Heavy Crude Oil Emulsion” by Mahmood M. S. Abdullah and co-workers presents the synthesis of 2 ionic liquids (ATEI-IL and ATHI-IL) and their application for chemical demulsification. To characterize the discussed system, the authors used several techniques, including FTIR and 1H-NMR spectroscopy, TGA and DTA analysis, surface tension measurement and optical microscopy.

I have the following comments:

  1. Page 2, line 81: Should be “carbonyl stretching band” instead of “carbonyl starching band”

Answer: It was modified to “carbonyl stretching bands”.

  1. Figure 1b: there is no wavelength value for the CH2 antisymmetric stretching vibration.

Answer: The wavelength value for the CH2 asymmetric stretching vibration was added to the figure.

  1. Page 5, line 119: the authors wrote that the CMC of ionic liquids was calculated “from the intersection between the regression straight line of the linearly dependent region and the straight line passing through the plateau” and indeed, it should be shown in Figure 4. Simple point joining is not appropriate in this case.

Answer: The figure was replaced to show the intersection between the regression straight line of the linearly dependent region and the straight line passing through the plateau.

  1. Page 6, line 129: “that” instead of “That”.

Answer: The capital letter was replaced with the small one.

  1. Figure 4: description of the x axis: should be “ln c”, instead of “lnC”

Answer: It was corrected.

  1. Figure 6 is not referenced anywhere.

Answer: Figure 6 was already referenced in the original manuscript in page 8 line 173

  1. The purity of the chemicals used should be reported.

Answer: The purity of the chemicals used was reported

  1. Scheme I, the last step of the synthesis: is “RBr” should be “RI”.

Answer: Scheme I, the last step of the synthesis: is “RBr” was replaced with “RI”.

  1. Chapter 3.4: the experimental details are poorly described.

Answer: Chapter 3.4: the experimental details were improved

  1. Page 12: the first sentence of the conclusion must be corrected.

Answer: The first sentence of the conclusion was corrected

  1. The practical relevance of the conclusions needs to be addressed.

Answer: The practical relevance of the conclusions was addressed

In my opinion, the manuscript is suitable for publication after making corrections.

Reviewer 3 Report

The manuscript is aimed at investigating the performance of synthetic amphiphilic ATEI-IL, and ATHI-IL for de-emulsification of W/O emulsions with different crude oil : brine volumetric ratios. The key results are the obtained data related to optimization of the de-emulsifier formulation which might be useful for further industrial application.

The investigation is of interest to the audience of Molecules. Aside from just reporting the results, it would be important to add some clarification about the basic phenomena that stand behind the study and the obtained results. There are also some unclear statements that should be rephrased or explained:

  1. lines 151-152: “AILs contain hydrophobic and hydrophilic moieties which means they can serve as nonionic surfactants.“ What does ‘serve as nonionics’ mean?
  2. line 153: “AILs show high toleration.” is an unclear statement.
  3. lines 194-195: What does ‘soft films’ mean? What is the definition of softness, as related to coalescence?
  4. line 201: What is ‘clean water’?
  5. lines 274-275: ‘was reacted’ are unclear statements.
  6. lines 285-287 “Increasing the hydrophobicity of AIL facilitates their diffusion in the crude oil as a continuous phase and thus prompts their arrival to the water/oil interface to reduce IFT and thereby replace the asphaltenes interfacial film.” This is completely imprecise statement. What is actually the proposed mechanism of replacement of asphaltenes?

Adding a short list of the abbreviations used might be useful, particularly for wider audiences. Otherwise the reader has to go every time through the text, so as to identify their meaning.

Author Response

Reviewer 3

The manuscript is aimed at investigating the performance of synthetic amphiphilic ATEI-IL, and ATHI-IL for de-emulsification of W/O emulsions with different crude oil : brine volumetric ratios. The key results are the obtained data related to optimization of the de-emulsifier formulation which might be useful for further industrial application.

The investigation is of interest to the audience of Molecules. Aside from just reporting the results, it would be important to add some clarification about the basic phenomena that stand behind the study and the obtained results. There are also some unclear statements that should be rephrased or explained:

  1. lines 151-152: “AILs contain hydrophobic and hydrophilic moieties which means they can serve as nonionic surfactants.“ What does ‘serve as nonionics’ mean?

Answer: Serve as nonionic surfactants means that they can behave and work as nonionic surfactants. The sentence was updated to be:

 “AILs contain hydrophobic and hydrophilic moieties which means they can behave and work as nonionic surfactants.”  

  1. line 153: “AILs show high toleration.” is an unclear statement.

Answer:  We replaced toleration with performance to enhance the statement.

  1. lines 194-195: What does ‘soft films’ mean? What is the definition of softness, as related to coalescence?

Answer:  Soft films are flexible films where the as-synthesized AILs replace the interfacial asphaltene rigid film with the soft one. The produced soft film can easily rupture to release water facilitating the coalescence process.

  1. line 201: What is ‘clean water’?

Answer: the clean water means that the clarity of the produced demulsified water after the demulsification process, where this water does not contain large amount of crude oil. As the reviewer can notice from Figure 7, both AILs succeeded to demulsify clear water. The clarity of demulsified water with the usage of both AIL can be seen easily through naked-eye observation in Fig. 7

  1. lines 274-275: ‘was reacted’ are unclear statements.

Answer: the statements were clarified

  1. lines 285-287 “Increasing the hydrophobicity of AIL facilitates their diffusion in the crude oil as a continuous phase and thus prompts their arrival to the water/oil interface to reduce IFT and thereby replace the asphaltenes interfacial film.” This is completely imprecise statement. What is actually the proposed mechanism of replacement of asphaltenes?

Answer: the proposed mechanism was added to the Results and Discussion section, as well as to the conclusion

Adding a short list of the abbreviations used might be useful, particularly for wider audiences. Otherwise the reader has to go every time through the text, so as to identify their meaning.

Answer: A short list of the abbreviations was added to the manuscript

Round 2

Reviewer 1 Report

There are still some problems that need to be further considered before publication.

  1. L190, “By increasing the ATHI-IL concentration in these emulsions, the hydrophobic moieties of the excess molecules can be oriented in the opposite direction at the water/oil interface. Therefore, the hydrophilic chains of ATHI-IL and other water droplets are bonded together, inhibiting the demulsification process.”

ATHI-IL are evenly dispersed in the continuous oil phase before the demulsification. All of them inevitably transfer to the oil-water interface due to lower interfacial intension and high interfacial activity. In other word, ATHI-IL are evenly distributed on the surface of all water droplets and the hydrophilic end enters into the water phase. How does ATHI-IL catch other droplets by the hydrophilic chains with the further increase of the concentration. Whether a new ATHI-IL film is formed when excessive demulsifer is added.

  1. L203, “AIL displayed a different type of interaction from traditional surfactant, such as ionic in-teraction between the AIL ions and the opposite charge of salt's ions in brine. These interactions reduce the repulsion between molecules of AIL at the W/O interface, allowing them to accumulate this interface. Due to these interactions, the IFT at the W/O interface is reduced and the asphaltene interfacial rigid film is weakened, making it easier to replace and, therefore, less stable for emulsion droplets [16].”

The explanation suggests that the salinity reduces the stability of emulsion and improves the demulsification. However, lower demulsification efficiency can be obtained when the brine content is relatively high. Obviously, the description doesn’t explain the result of the experiments.

Author Response

Comments and Suggestions for Authors

There are still some problems that need to be further considered before publication.

  1. L190, “By increasing the ATHI-IL concentration in these emulsions, the hydrophobic moieties of the excess molecules can be oriented in the opposite direction at the water/oil interface. Therefore, the hydrophilic chains of ATHI-IL and other water droplets are bonded together, inhibiting the demulsification process.”

ATHI-IL are evenly dispersed in the continuous oil phase before the demulsification. All of them inevitably transfer to the oil-water interface due to lower interfacial intension and high interfacial activity. In other word, ATHI-IL are evenly distributed on the surface of all water droplets and the hydrophilic end enters into the water phase. How does ATHI-IL catch other droplets by the hydrophilic chains with the further increase of the concentration. Whether a new ATHI-IL film is formed when excessive demulsifer is added.

Answer: It was corrected to ‘’ That can be attributed to the aggregation of elevated concentrations of IL molecules at water/oil interface to form new ATHI-IL films around water droplets which lead to more emulsification instead of demulsification’’.

  1. L203, “AIL displayed a different type of interaction from traditional surfactant, such as ionic in-teraction between the AIL ions and the opposite charge of salt's ions in brine. These interactions reduce the repulsion between molecules of AIL at the W/O interface, allowing them to accumulate this interface. Due to these interactions, the IFT at the W/O interface is reduced and the asphaltene interfacial rigid film is weakened, making it easier to replace and, therefore, less stable for emulsion droplets [16].”

The explanation suggests that the salinity reduces the stability of emulsion and improves the demulsification. However, lower demulsification efficiency can be obtained when the brine content is relatively high. Obviously, the description doesn’t explain the result of the experiments.

Answer: This statement was written in a confusing way and it was corrected; what we meant is that there is a difference between traditional surfactants that, in sometimes, lose their activity in high salinity conditions and AIL that can disperse easily under the same conditions of salinity. Moreover, they interact with the oppositely charged salt ions that help in reducing the repulsion between ionic liquid molecules leading to more accumulation at W/O interface. This high accumulation helps to enhance their demulsification activity by increase their availability at the W/O interface.

However, the high salinity of sea water, when forming the emulsion and before adding the demulsifier, plays an important role in enhancing the stability of water/oil emulsion. 

Comments and Suggestions for Authors

There are still some problems that need to be further considered before publication.

  1. L190, “By increasing the ATHI-IL concentration in these emulsions, the hydrophobic moieties of the excess molecules can be oriented in the opposite direction at the water/oil interface. Therefore, the hydrophilic chains of ATHI-IL and other water droplets are bonded together, inhibiting the demulsification process.”

ATHI-IL are evenly dispersed in the continuous oil phase before the demulsification. All of them inevitably transfer to the oil-water interface due to lower interfacial intension and high interfacial activity. In other word, ATHI-IL are evenly distributed on the surface of all water droplets and the hydrophilic end enters into the water phase. How does ATHI-IL catch other droplets by the hydrophilic chains with the further increase of the concentration. Whether a new ATHI-IL film is formed when excessive demulsifer is added.

Answer: It was corrected to ‘’ That can be attributed to the aggregation of elevated concentrations of IL molecules at water/oil interface to form new ATHI-IL films around water droplets which lead to more emulsification instead of demulsification’’.

  1. L203, “AIL displayed a different type of interaction from traditional surfactant, such as ionic in-teraction between the AIL ions and the opposite charge of salt's ions in brine. These interactions reduce the repulsion between molecules of AIL at the W/O interface, allowing them to accumulate this interface. Due to these interactions, the IFT at the W/O interface is reduced and the asphaltene interfacial rigid film is weakened, making it easier to replace and, therefore, less stable for emulsion droplets [16].”

The explanation suggests that the salinity reduces the stability of emulsion and improves the demulsification. However, lower demulsification efficiency can be obtained when the brine content is relatively high. Obviously, the description doesn’t explain the result of the experiments.

Answer: This statement was written in a confusing way and it was corrected; what we meant is that there is a difference between traditional surfactants that, in sometimes, lose their activity in high salinity conditions and AIL that can disperse easily under the same conditions of salinity. Moreover, they interact with the oppositely charged salt ions that help in reducing the repulsion between ionic liquid molecules leading to more accumulation at W/O interface. This high accumulation helps to enhance their demulsification activity by increase their availability at the W/O interface.

However, the high salinity of sea water, when forming the emulsion and before adding the demulsifier, plays an important role in enhancing the stability of water/oil emulsion.